# Local Antibiotic Delivery Options in Prosthetic Joint Infection

**DOI:** 10.3390/antibiotics12040752

**Published:** 2023-04-14

**Authors:** William Steadman, Paul R. Chapman, Michael Schuetz, Beat Schmutz, Andrej Trampuz, Kevin Tetsworth

**Affiliations:** 1Jamieson Trauma Institute, Royal Brisbane and Women’s Hospital, Herston, Brisbane 4029, Australia; 2Department of Orthopaedics, Royal Brisbane and Women’s Hospital, Herston, Brisbane 4029, Australia; 3Faculty of Health, Queensland University of Technology, Brisbane 4059, Australia; 4Herston Infectious Disease Institute, Royal Brisbane and Women’s Hospital, Herston, Brisbane 4029, Australia; 5Department of Infectious Diseases, Royal Brisbane and Women’s Hospital, Herston, Brisbane 4029, Australia; 6School of Mechanical, Medical and Process Engineering, Faculty of Engineering, Queensland University of Technology, Brisbane 4000, Australia; 7Centre for Biomedical Technologies, Queensland University of Technology, Brisbane 4059, Australia; 8Australian Research Council Training Centre for Multiscale 3D Imaging, Modelling, and Manufacturing, Queensland University of Technology, Brisbane 4059, Australia; 9Center for Musculoskeletal Surgery, Septic Unit Charité-Universitätsmedizin, 10117 Berlin, Germany; 10School of Medicine, University of Queensland, Brisbane 4029, Australia

**Keywords:** prosthetic joint infection, orthopaedic device related infection, local antibiotics, exchange arthroplasty, revision arthroplasty, calcium sulphate, calcium phosphate, bioactive glass, hydrogel, bacteriophages, debridement, antibiotics and implant retention, DAIR

## Abstract

Prosthetic Joint Infection (PJI) causes significant morbidity and mortality for patients globally. Delivery of antibiotics to the site of infection has potential to improve the treatment outcomes and enhance biofilm eradication. These antibiotics can be delivered using an intra-articular catheter or combined with a carrier substance to enhance pharmacokinetic properties. Carrier options include non-resorbable polymethylmethacrylate (PMMA) bone cement and resorbable calcium sulphate, hydroxyapatite, bioactive glass, and hydrogels. PMMA allows for creation of structural spacers used in multi-stage revision procedures, however it requires subsequent removal and antibiotic compatibility and the levels delivered are variable. Calcium sulphate is the most researched resorbable carrier in PJI, but is associated with wound leakage and hypercalcaemia, and clinical evidence for its effectiveness remains at the early stage. Hydrogels provide a versatile combability with antibiotics and adjustable elution profiles, but clinical usage is currently limited. Novel anti-biofilm therapies include bacteriophages which have been used successfully in small case series.

## 1. Introduction

Prosthetic joint infection (PJI) has been a battle for the medical profession since prosthetic joints began to be implanted, and rates vary between 0.85% and 2% worldwide [1,2]. In Australia, there are estimated to be 3900 new PJI cases each year [3]. The number of revisions for PJI is increasing as both the rate of primary joint implantation increases with ongoing demand, as well as an increasing revision rate for infection reported in the United States and United Kingdom [2,4,5].

PJI carries a significant burden for patients and healthcare systems alike. For PJI patients, treatment is prolonged and arduous, with psychosocial stress comparable to oncology patients [6]. Long-term functional scores are poorer than the general population despite successful treatment, with even worse outcomes for those requiring multiple re-operations [7]. There is a significant mortality risk, with systematic reviews demonstrating over 4% 1-year mortality and over 20% 5-year mortality for patients diagnosed with hip or knee PJI, across over 19,000 and 20,000 patients, respectively [8,9]. Treatment is costly for healthcare systems, with revision surgery for infection costing significantly more than aseptic causes, predominantly due to greater peri-operative costs, prosthesis costs, and hospital stay [10,11,12,13,14,15]. A recent meta-analysis of the cost of septic revision knee arthroplasty found it to be double the cost of aseptic revision, up to USD 38,109, and two-stage septic revision with one re-revision costing as much as USD 81,938 [10].

Conventional treatment requires surgery and includes Single Stage Exchange arthroplasty (SSE), Two Stage Exchange arthroplasty (TSE) or Debridement, Antibiotics, and Implant Retention procedures (DAIR). This usually occurs in conjunction with long-term targeted systemic antibiotic therapy. Local administration of antibiotics aims to improve treatment success, limit antibiotic-associated adverse outcomes, and avoid antimicrobial resistance through reduced systemic exposure to antimicrobials. The goal of this review is to discuss administration of local antimicrobial agents and their carriers, and review current evidence in relation to PJI.

## 2. Biofilm Pathology

The development of biofilm is thought to be the principal microbial factor leading to the observed outcomes of PJI management, and orthopaedic device related infection (ODRI) generally. Bacteria in an established biofilm are up to 10,000 times more resistant to antibiotic therapy than their free-floating planktonic counterparts [16]. Irreversible attachment to the surface begins within one minute of initial attachment, increasing by 100 times over ten minutes [17]. This biofilm forms on both artificial surfaces and local tissues.

The multi-layered structure of the biofilm, as seen in Figure 1, provides resistance to immune system clearance and antibiotic penetration. It consists of a more metabolically active outer layer that promotes further bacterial dissemination through release of planktonic variants, and an inner layer with reduced activity [18]. These cells may form a nidus persisting through antimicrobial exposure (known as ‘persister cells’), resulting in relapse after treatment [18,19,20]. The extra-cellular polymeric substances (EPS) matrix surrounding cells in the mature biofilm also limits penetration of antibiotic compounds and diminishes the effectiveness of the immune system [21]. The EPS protects bacterial cells from leukocyte-based killing of bacteria and reduces immune-cell phagocytosis [22,23,24,25,26]. It also reduces complement activation, and affects polymorphonuclear leukocyte bactericidal ability [27]. Horizontal plasmid-based gene transfer of antibiotic resistance has been demonstrated in biofilms [28]. The biofilm phase may increase genetic expression of secreted bacterial protective factors, including multidrug efflux pumps and stress response regulons [18,29]. This altered state of genetic expression compared to the planktonic form also exhibits increased resistance to bactericidal compounds [30].

Successful methods of biofilm eradication therapy may result in increased treatment success and reduced procedural morbidity and complexity. Delivering local antibiotic therapy directly to the site of infection carries significant potential to achieve this goal. With greater biofilm destruction, it may improve success rates of current treatment strategies, broaden indications for less invasive surgical management, or in some cases may avoid surgery altogether. This has the potential to enhance the patient treatment experience, improve functional outcomes, and reduce the financial cost of PJI treatment.

## 3. Local Antibiotics

Placing antibiotics directly into the site of infection carries several advantages over systemic administration. It can provide significantly higher concentrations to infected tissue with reduced systemic exposure. This may lower rates of antibiotic-associated adverse events and decrease antimicrobial resistance through preservation of the gastrointestinal microbiome.

Local antibiotics can be administered ‘naked’, or with a carrier to improve antibiotic elution. Antibiotic carriers include polymethylmethacrylate bone cement, calcium sulphate, calcium hydroxyapatite, bioactive glass, and hydrogels. Non-antibiotic options, such as bacteriophages and ceragenins, aim to destroy biofilm more efficiently but are still in initial stages of clinical development.

### 3.1. Antibiotic Choice

Choosing an antibiotic to be used locally requires consideration of several factors. An antibiotic can be selected either to target a known organism, or empirically to target a spectrum of common PJI pathogens. Antibiotics may be toxic to bone and soft tissues in very high concentrations, but fluoroquinolones, tetracyclines, and rifampicin have been shown to be particularly toxic to osteogenic cells even at low levels [31]. As with all antibiotic usage, prudent stewardship is extremely important to protect against development of antimicrobial resistance [32]. Patients may be at risk of systemic adverse effects from an antibiotic, such as kidney injury or allergy [33]. Finally, if a carrier is used, the interaction between the antibiotic and the carrier also needs to be considered. Selection of an appropriate antibiotic requires consideration of these infection, antibiotic, and host factors on a case-by-case basis.

### 3.2. Pharmacokinetics

Ideal local antibiotic administration provides predictable, high concentrations in surrounding tissue with minimal systemic exposure or adverse events. Importantly, although antibiotic efficacy determinants of microbial killing are defined in planktonic form, determinants of antibiotic efficacy in local administration have not yet been fully determined.

Antimicrobial susceptibility of bacteria in biofilm may be assessed by the Minimum Biofilm Eradication Concentration (MBEC), which can be multiple orders of magnitude higher than the Minimum Inhibitory Concentration (MIC) of planktonic bacteria. For example, vancomycin-sensitive *Staphylococcus aureus* may have an MBEC of over 1000 μg/mL compared to its planktonic MIC of 2 μg/mL. Systemic therapy is unable to reach these levels safely [34].

### 3.3. Naked Antibiotics

Directly placing antibiotics into the surgical site as powder or solution has been studied as PJI prophylaxis in primary joint replacement. Studies are limited with no randomised controlled trials (RCT), and results of recent meta-analyses are conflicting as to whether PJI risk is reduced [35,36]. Vancomycin powder has been shown to significantly reduce the gram-positive deep surgical site infection rate in an RCT of high-risk tibial articular fractures [37]. It has also been shown to be protective against surgical site infection in spinal surgery on meta-analysis [38]. In primary total joints, antibiotic levels in drain output of patients who received intra-articular vancomycin powder have been measured. Local levels were highest at first measurement three hours post-operatively, but rapidly declined with a half-life of 7.2 h and were estimated to be subtherapeutic by 64 h post-op; however, the minimum duration to achieve clinical effect is unknown [39].

### 3.4. Regular Intra-Articular Administration

The regular administration of antibiotics directly into the joint is well established in large animal veterinary science [40]. Whiteside and colleagues have published several case series using a treatment protocol for hip and knee PJI, intra-operatively placing a Hickman catheter inside the joint at the time of revision surgery and administering vancomycin or gentamicin. In these case series of 18 to 30 patients, there are impressive infection success rates for high-risk patients undergoing SSE for recurrent knee PJI (94%) or for methicillin-resistant *Staphylococcus aureus* (MRSA) (94%). The authors encountered several adverse events, mostly catheter-related issues, antibiotic precipitation, or renal dysfunction related to antibiotic therapy (20%) [41,42,43]. Several other authors have published case series of their varied protocols with comparable outcomes in culture negative PJI and fungal PJI [44,45].

Roy et al. treated twelve knee PJIs with intraarticular vancomycin for six weeks and assessed peak and trough synovial vancomycin levels. Participants received between 300–500 mg twice daily which was decreased if serum levels exceeded 20 μg/mL. Peak concentrations reached a mean of over 9000 μg/mL, well above the MBEC for common ODRI pathogens, with a mean trough of 377 μg/mL [34]. However, half-life was between 1.6 and 4.7 h, and there was high variability in intra-articular concentrations between patients, up to a factor of three [46]. An animal model of MRSA PJI has also shown a reduction in periprosthetic biofilm, bone loss, and local soft tissue reaction with administration of intra-articular antibiotics, compared to systemic administration [47].

## 4. Local Antibiotic Carriers

Use of an antibiotic carrier is an alternative method of maintaining high local concentrations of antibiotic without systemic exposure. These substances incorporate an antimicrobial to prolong its half-life and provide predictable elution characteristics. They may also serve additional roles, such as filling dead space and providing mechanical support for limb alignment. The ideal antibiotic carrier would provide prolonged antibiotic concentrations at an effective level and achieve complete antibiotic release to minimise subtherapeutic elution time. It would also be versatile and have compatibility with the desired antibiotics, and fully resorbable with minimal risk of allergies, and systemic or local adverse effects.

### 4.1. PMMA

Polymethylmethacrylate (PMMA) bone cement has a long history of use as an antibiotic carrier, with Buchholz and Engelbrecht first publishing on antibiotics mixed with Palacos cement in 1970 [48]. Structural cement spacers that bear load are an important part of staged revision for PJI. The spacer is inserted following debridement at the first stage of TSE to allow weight-bearing, maintain bony alignment, and aid in mobility and pain management. Local antibiotic elution comes from mixing antibiotics with PMMA before curing, and is a balance between increasing antibiotic concentration and decreasing structural integrity. Antibiotics are available pre-mixed from multiple manufacturers, or may be mixed at the time of surgery. It is effective in reducing infection recurrence, and the use of an antibiotic loaded spacer in TSE is associated with a 7% absolute risk reduction [49]. Structural spacers have lower antibiotic elution when compared to beads of the same composition [50]. This is due to beads having greater surface area to volume ratio, with surface elution from PMMA responsible for initial burst levels [51]. Following this, slower elution is through the network of interconnecting pores and cracks in the material, and passive diffusion through the cement matrix [52]. Antibiotic beads have been shown to be non-inferior to conventional intravenous antibiotic therapy in osteomyelitis treatment [53,54]. In a randomised controlled trial of 28 hip and knee PJI patients, antibiotic beads had a lower infection recurrence rate (15%) compared to conventional parenteral therapy (30%) following prosthesis resection, although this was not statistically significant [55].

Effective local therapy has questioned the need for prolonged systemic antibiotic treatment in some studies. Stockley et al. provided targeted local antibiotic therapy with PMMA beads at the first stage of 114 TSE for chronic hip PJI [56]. Patients only received three doses of systemic peri-operative antibiotic prophylaxis, with a success rate of 87.7% at a minimum two-year follow-up. From the same centre, a case series of 53 chronic knee PJIs undergoing TSE or staged arthrodesis reported an infection eradication rate of 89% with use of targeted antibiotics in PMMA beads and spacer [57]. These patients received a mean of only 4.6 days of intravenous therapy, and only three patients received subsequent oral antibiotics. Hart et al. reported on 48 TSEs for knee PJI using PMMA spacers containing gentamicin and vancomycin, with 88% infection eradication following only two weeks of post-operative intravenous therapy after the first stage [58].

Variable antibiotic pharmacokinetics have been reported when using PMMA as an antibiotic carrier. Local antibiotic level peaks between day one and three, indicating an initial burst-release followed by a slow decline in concentration; however, the peak levels and length of time above the MIC is variable between studies and difficult to control [59,60]. Variability is due to multiple factors, including antibiotic type and concentration, PMMA type, mixing technique, physical wear, use of prefabricated or handmade products, and anatomic location with local fluid turnover [51,61]. Nonetheless, peak local antibiotic levels vastly exceed those achievable with systemic therapy [46]. PMMA antibiotic release is poor, with only 25–50% of contained antibiotic eluted in bead form [53]. Shaping smaller beads with greater surface area can improve elution, with 3 × 5 mm ‘mini-beads’ releasing 93% of contained antibiotics over 14 days and achieving local concentrations seven times higher than 7 × 7 mm beads [62].

In PJI, studies that measure antibiotic release from PMMA are predominantly in structural spacers. Some studies demonstrate several months of continuous antibiotic release, while others reveal that local concentrations drop below MIC after seven to fourteen days [54,63]. A systematic review of *in-vivo* cement spacer antibiotic levels by Anagnostakos and Meyer found no clearly superior cement or antibiotic mix with significant heterogeneity of studies in cement spacer composition and antibiotic level sampling technique [59]. Invitro studies to define the superior composition of cement in spacers unfortunately have conflicting results. Palacos with gentamicin has shown highest elution in initial studies; however, recent literature has shown other cements to have similar elution kinetics, with variable superiority when changing or adding impregnated antibiotics [59,60,63,64,65,66,67,68]. Antibiotic elution may be increased through altering composition by increasing antibiotic concentration or putting other additives in the cement such as glycine, Poly(lactic-co-glycolic) Acid (PLGA), or calcium phosphate [51,63,69]. Stevens et al. demonstrated that spacers with higher antibiotic doses achieved higher burst levels and eluted antibiotics above the MIC for over 80 days [63].

Mixing technique can alter cement porosity, which subsequently affects antibiotic release. Hand mixing, in comparison to vacuum mixing, can incorporate more air which can increase peak antibiotic concentration up to five-fold in vitro [65,70]. Vacuum mixing has a variable effect on antibiotic elution, and in vitro studies confirm improved antimicrobial activity with Cobalt G-HV, Palacos R+G, and Simplex P, while declining performance when using Cemex Genta, Smartset GMV, and Versabond AB [71]. Using multiple antibiotics can similarly increase porosity and synergistically improve release of both antibiotics [70]. However, contradicting studies have also found no difference between mixing techniques or with the use of multiple antibiotics [72,73]. 

PMMA may be incompatible with antibiotics due to its effects on the antibiotics themselves, or vice versa. The exothermic polymerisation reaction of PMMA can generate temperatures up to 90 degrees Celsius [51]. This may degrade contained antibiotics, and heat stability should be considered in antibiotic choice. In vitro, beta-lactams are highly fragile, while gentamicin has only a slight decrease in activity after heat treatment. Other antibiotics including aminoglycosides, glycopeptides, tetracyclines and quinolones are stable over six weeks following initial heat treatment [74]. Mechanical changes can also occur in PMMA induced by some antibiotics. Antibiotics in liquid form have a greater impact on the structural integrity of cement compared to powdered equivalents, and some antibiotics such as rifampicin cause significant changes in cement consistency during polymerisation resulting in decreased mechanical strength [51].

Local adverse effects of PMMA appear to be minimal once the cement is cured and the risk of thermal injury has passed. However systemic adverse effects due to supratherapeutic antibiotic levels are being increasingly recognised, especially aminoglycoside nephrotoxicity. Serum levels of gentamicin and vancomycin from spacers are detectable for up to eight weeks [75], and spacers with high dose vancomycin or aminoglycoside (over 3.6 g antibiotic per 40 g PMMA) have been associated with almost two-fold increase in acute kidney injury (AKI) risk compared to lower dose spacers [33]. In two systematic reviews, overall rate of AKI following TSE with an antibiotic spacer is 4.18%, compared to 0.55% in primary arthroplasty [49,76]. However, this may simply reflect intraoperative hypovolemia and the published studies are clearly susceptible to multiple biases, primarily selection bias. These revision cases are of course far more complicated and prolonged than standard primary arthroplasty, and these results must be interpreted carefully and viewed with caution.

Mechanical studies have demonstrated that PMMA does not appear to lose strength due to elution of antibiotics; however, persistence of the carrier has been associated with the emergence of antibiotic resistance [77]. Small colony variants emerge following prolonged exposure to subtherapeutic levels of antibiotic, and resistant organisms have been isolated in retrieved cement spacers, although the incidence is unclear [78]. Persistent cement may also be a focus for clinical infection recurrence [79]. Removal of PMMA generally requires a second surgery, which carries the risk of local or systemic complications.

Regardless, PMMA remains an essential antibiotic carrier in the treatment of PJI and is by far the most studied. It is well understood as a biomaterial in orthopaedic surgery due to its long history of use and has effective elution of local antibiotics with proven reduction in infection recurrence in clinical studies. Its ability to bear load is not shared by any other available antibiotic carrier. It is not without issues, with variability in antibiotic levels, incompatibility with several classes of antibiotics, and the requirement for removal following infection clearance. Finally, antibiotic elution from PMMA, despite much lower systemic levels than parenteral therapy, has still been associated with occasional systemic adverse effects. Figure 2 illustrates several evidence-based recommendations from the authors to improve elution characteristics. 

### 4.2. Resorbable Carriers

Resorbable carriers aim to completely degrade to overcome the persistence issues of PMMA and do not require a second surgery for removal. This avoids subtherapeutic levels associated with carrier persistence in PMMA, as all antibiotics are released when the carrier is fully resorbed. In PJI, their resorbable nature makes them more attractive in DAIR or SSE, when there is no planned return to theatre to facilitate removal, as in TSE. Cements or gels may also be mouldable to fit defects, manage surgical dead space and increase antibiotic penetrance.

### 4.3. Calcium Sulphate

Calcium sulphate (CS) may be used as an antibiotic carrier, and its effectiveness in chronic osteomyelitis and fracture-related infection has led to further investigation in PJI [53,80]. It is promising as a bioabsorbable antibiotic carrier with favourable elution qualities, although it carries a risk of persistent wound leakage, heterotrophic ossification, or life-threatening hypercalcaemia [81]. It is available as cement which can be shaped into beads to suit a variety of clinical applications (Figure 3). It releases 100% of loaded antibiotics over time, and elutes over a period of several weeks [82]. When combined with targeted antibiotics, it has been reported to successfully inhibit biofilm formation and eradicate established biofilms in vitro in multiple bacterial species [83]. It has also been shown in vitro to cause minor third-body wear when used in prosthetic joints; however, rates are much lower than with PMMA or ceramic particles [84]. A systematic review of wound leakage in prosthetic joint surgeries using CS beads revealed a rate of 3.8%, even with deep usage inside the joint, and the included larger studies suggested that higher volumes of CS or medical co-morbidities posed an increased risk of this complication [81,85,86]. A separate systematic review of hypercalcaemia demonstrated a 4.2% overall rate; however, only 0.28% required management, with one case out of 1049 being life-threatening and needing intensive care [87]. Unfortunately, the comparative studies assessing success rates are limited to case series or case-control studies in DAIR procedures. A matched case-control study of 40 DAIRs for acute hip or knee PJI demonstrated no significant improvement with the use of vancomycin and tobramycin CS beads at 90 days or two years [88]. This study reported a 45% failure rate, consistent with other smaller published case series of CS DAIRs [89]. However, a recently published cohort study by Reinisch et al. identified significant improvement in 41 DAIRs for hip PJI, with vancomycin, ceftriaxone, or tobramycin CS beads [90]. The CS group had a significantly lower revision rate of 15% compared to 64% in the standard group. Studies of CS in SSE or TSE for PJI either do not have re-infection rate as their primary outcome, have no comparator group, or have grouped aseptic and septic revision. Reported re-infection rates range from 0% to 6.7% [85,86,91]. While further higher-quality studies need to be performed, the significant cost and risk of adverse effects of CS carriers should be considered before recommending their routine clinical use in the absence of any demonstrated benefit.

### 4.4. Hydroxyapatite

Calcium hydroxyapatite (CHA) has seen recent renewed interest as an antibiotic carrier, despite first publication of its use over 20 years ago [92]. A number of calcium apatite compounds are available, including hydroxyapatite and tricalcium phosphate [92]. These can be prepared as blocks, with an encased antibiotic powder reservoir, or as a cement mixed with antibiotic powder. Isothermic hardening of cements has the advantage of avoiding thermal damage to antibiotics, and its porous nature promotes biological reactivity for bone formation and antibiotic release [92,93]. However, the slow and incomplete degradation of these cements causes incomplete and inconsistent antibiotic elution, and promotes bacterial colonisation [92]. A case series of fourteen knee PJI patients with antibiotic-impregnated CHA pellets (Bone Ceram P, Olympus Terumo Biomaterials Corp, Tokyo, Japan) at the first stage of TSE reported a recurrence rate of 21% at mean follow-up of 5.1 years [94]. They reported no complications related to use of CHA. However, inconsistent antibiotic elution needs to be addressed before it can be used as a reliable carrier.

Combinations of CHA and CS are more clinically studied in bone and joint infection. Cerament® (BoneSupport AB, Lund, Sweden) is a cement mixture of 40% CHA and 60% CS, and is commercially available with gentamicin or vancomycin (Figure 4) [95]. In vitro, the substance has been shown to increase prosthesis pull-out strength and to inhibit biofilm formation with antibiotics [96,97]. When used clinically in PJI patients, it can deliver high local antibiotic levels, with drain effluent levels well above MIC at mean 90 h post-op [98]. However, this study also noted that presence of a post-operative drain significantly reduces available local antibiotic by measuring renal antibiotic clearance. Logoluso et al. found a 95% infection eradication rate with targeted antibiotic-laden Cerament® applied to stemmed components at second-stage of TSE for PJI [99]. Across two large case series of chronic osteomyelitis, including 71% following infected fracture fixation, McNally et al. reported 94% infection eradication at four to eight years follow-up in higher risk Cierny–Mader type B hosts [95,100]. PerOssal® (Osartis GmBH, Dieburg, Germany), a mix of 51.5% nanocrystalline CHA and 48.5% CS, has been used in PJI at first-stage of TSE combined with antibiotics within the intramedullary canal [101]. The PerOssal group reported a trend towards improved infection recurrence rate (6.67% vs. 16.13%), and significantly lower serum inflammatory marker values from the second to sixth postoperative weeks. There was no difference in complication rate between groups, although they did note delayed wound healing when the substance was used outside of the intramedullary canal. Overall, this combination carrier has significant potential in PJI given its promising results in osteomyelitis; however, similarly to CS, it must be used with caution to avoid local wound healing complications.

### 4.5. Bioactive Glass

Bioactive glass (BAG) carries inherent antibacterial activity following implantation and can also act as an antibiotic carrier. It is an attractive option in osteomyelitis and fracture-related infection due to stimulation of bone formation to heal defects, and has been shown to improve local vascularity with proangiogenic properties [102]. Once implanted, calcium is released into the local environment from the glass, which reacts with interstitial phosphate to form a calcium phosphate layer on the surface of the BAG. Ongoing growth of this layer continues with further dissolution of the glass, which then undergoes crystallisation to hydroxyapatite allowing cellular adhesion [102,103,104]. Sodium and potassium are also released from the substance and exchanged for ionic hydrogen. This increases osmotic pressure and local pH up to 11.65, creating a hostile environment for microbes [102,104]. BAG-S53P4 (commercially available as Bonalive®, Turku, Finland) is a silicate glass that provides bactericidal activity against a broad-spectrum of planktonic and biofilm-phase bacteria, including multi-resistant organisms, and also prevents biofilm formation [105,106,107]. In vitro bacterial killing has been shown to be equivalent to antibiotic-loaded PMMA [108]. Resistance to BAG has not been reported during in vitro studies [104]. The porous nature of mesoporous and sol-gel glasses allows for release of metal ions and larger molecules, including antibiotics. Metal ions can be incorporated at the time of manufacture and silver, gallium, magnesium, copper, strontium, and zinc have shown positive in vitro results [103,104,109]. Antibiotic compounds have been incorporated into BAG during in vitro studies, with total antibiotic release up to 90%; however, similar to PMMA, heterogeneity exists in both BAG composition and antibiotic used, leading to variability in the level released and the duration [110,111,112,113,114]. Newer composite borate glasses exhibit promising elution kinetics compared to silicate glasses and possess tailorable elution qualities based on their composition [115]. In animal models, antibiotic-loaded borate glass revealed significant improvement in infection clearance and histological tissue quality over glass alone, suggesting that antibiotic therapy provides extra antibacterial activity [111,112,113,115,116]. Antibiotic release has been detected in rabbit models out to 21 days for teicoplanin, and fourteen days for vancomycin [111,112].

There are no published studies of clinical use of BAG in prosthetic joint infection currently, but non-antibiotic-laden BAG has been used in osteomyelitis treatment following surgical debridement. Lindfors et al. reported an overall cure rate of 89.7% at 31 months in a multinational register of 116 patients, predominantly involving post-traumatic long bone osteomyelitis [117]. In this study 84.5% of patients received BAG-S53P4 without antibiotics as a single-stage procedure, while 15.5% initially received antibiotic-loaded PMMA beads followed by BAG-S53P4 at the second stage. They also reported significantly more early poor outcomes in the two-stage treatment group. Another prospective study of BAG-S53P4 in 27 cases of long bone osteomyelitis found 88.9% of patients were infection-free at mean 17.8 months follow-up [106]. A comparative retrospective study of long bone osteomyelitis compared BAG-S53P4 to hydroxyapatite plus calcium sulphate plus antibiotics and tricalcium phosphate plus demineralised bone matrix plus antibiotics [118]. They found 92.6% of patients who received BAG-S53P4 were infection-free at mean 21.8 months with equivalent infection clearance across the three groups, but significantly lower wound complication rates in the BAG group. Two smaller retrospective case series of eleven and three patients reported infection clearance rates of 91% and 100% after treatment with BAG-S53P4 following debridement [102,119]. Importantly, all patients in all studies discussed received systemic antibiotic therapy. Despite these results in the treatment of osteomyelitis, further research is required to understand the utility of bioactive glass in prosthetic joint infection, including its effect on articulated surfaces. Newer composite and borate glasses to deliver antibiotics also have potential to augment the natural antibacterial activity of BAG.

### 4.6. Hydrogels

Hydrogels carry significant promise as a fully resorbable carrier that can be tailored to a specific clinical indication. They can be manufactured from multiple biodegradable polymers, with variable antibiotic release kinetics and degradation time of the gel [120,121]. Their consistency also makes them versatile for application to implant surfaces or fill dead space, as seen in Figure 4. They can also carry a wide variety of antimicrobial substances as, in contrast to PMMA, hydrogel can be mixed at room temperature, without thermal damage to the contained antimicrobial. They have been combined in experimental studies or case reports with a variety of antibiotics, antifungals, chitosan, and tyrosol [122,123,124]. The high water content inside the hydrogel matrix is also an appropriate environment for containing and releasing bacteriophages [125]. Animal models have found that loaded hydrogels are able to achieve the MBEC for multiple common ODRI pathogens [126]. The forefront of hydrogel technology has produced layered microparticles within a hydrogel to enable multiphase release, in a rabbit model releasing vancomycin and ceftazidime at stable therapeutic level out to 56 days, while simultaneously administering lidocaine for pain relief to 14 days [127].

Clinical studies are limited to the use of Defensive Antibacterial Coating (DAC®, Novagenit Srl, Mezzo Lombardo, Italy), which is the most studied and is already commercially available (Figure 4). DAC® is recommended as a prophylactic implant coating, as it changes the implant surface from hydrophobic to hydrophilic, to impair bacterial adhesion, but is completely resorbed in 72 h [128]. It can also carry antibiotics to improve its capacity for infection prevention or for PJI treatment. The largest clinical study in PJI treatment is a 1:1 case-control study of cementless TSE by Zagra of 54 patients. At 2.7 years mean follow-up, they reported four recurrences in the control group (14.8%), compared to none in the group which received DAC with targeted antibiotics at the second stage [129]. Another matched case-control study of 44 patients found equivalence in infection recurrence rate between TSE compared to SSE with DAC and antibiotics, with a significant reduction in hospital stay and antibiotic duration in the DAC group [130]. However, importantly, all the SSE patients in this study met their criteria for SSE, for which other authors have found equivalent results between SSE and TSE (no large soft tissue defect, previously identified and antibiotic sensitive pathogen) [131]. In the largest study of DAC use with antibiotics, an RCT of 380 primary and revision arthroplasty patients, the treatment group was not found to have any complications attributable to the hydrogel, with equivalent wound healing, lab biomarkers, and cementless implant osseointegration [132]. This is consistent with the other smaller clinical and safety studies that have not found any significant complications associated with DAC use. These promising early clinical results for DAC suggest that a dedicated hydrogel for PJI could safely deliver therapeutic levels over a prolonged period.

### 4.7. Nanocarriers

Nanocarriers are complex molecular moieties that respond to internal or external stimuli for activation. Self-targeting carriers can respond to local changes in infected tissue, such as pH, hypoxia, macrophage presence, increased reactive oxygen species, local ligands, bacterial enzymes, or increased local temperature [133]. Nanocarriers can also respond to external stimuli such as photothermal radiation or ultrasound waves [134,135]. These carriers, once activated, change their structure from a mobile hydrophilic compound, becoming hydrophobic and releasing their antimicrobial agent. This can be an antibiotic, antifungal, heavy metal nanoparticle, or other bacterio-toxic substance. They can also be designed to release their antimicrobial substance with enhanced pharmacokinetics compared to current options [133]. Self-propelled ‘Nanorobots’ can be guided using photothermal stimulation to release their antimicrobial substances into a target area [136].

**Figure 4 antibiotics-12-00752-f004:**
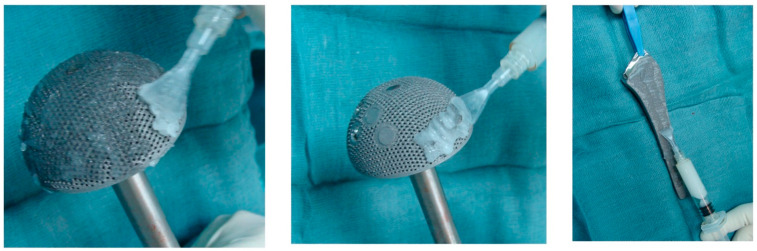
Defensive Antibacterial Coating (DAC) applied to acetabular and femoral components (from Romano et al. [132]).

## 5. The Future: Novel Antibiofilm Therapies

Novel therapies provide opportunities to disrupt biofilms and overcome increased antibiotic resistance with new mechanisms of action.

### 5.1. Bacteriophages

Bacteriophages (or phages) are naturally occurring bacterial viruses that are remarkably diverse and have co-evolved with bacteria throughout the history of life. Lytic phages undergo a life cycle of self-replication inside a bacterial cell following infection, before being released by lysing the bacteria. They were first recognised as a therapeutic option in 1917 but were quickly overtaken in the West by antibiotics. Development and clinical use did continue in Georgia and Russia, where phage cocktails are still available and clinically used. With a rise in antibiotic resistance, they have again become an area of interest in the West [137,138,139,140]. They have attractive inherent properties as they are generally specific to targeted bacteria, have no effect on human cells, can be trained to improve efficiency, and have naturally developed enzymes to actively lyse bacteria and break down EPS [141,142]. However, their clinical use in the West is in its early stages, with further development required to accumulate ‘phage libraries’ to target different bacterial species, resistance understanding, manufacturing techniques, regulation, and clinical administration [143,144,145].

Clinical use of phages is currently limited to salvage treatment for PJI [146]. Ferry et al. treated a case of recurrent methicillin-sensitive *Staphylococcus aureus* hip PJI, that later grew multidrug resistant (MDR) *P. aeruginosa.* Intra-articular phage cocktails were administered against both organisms, with additional targeted antibiotic therapy following a DAIR procedure. While eradication of these microbes was achieved, the patient remained on suppressive therapy for other identified bacteria, but achieved satisfactory function [138]. This group has also managed recurrent pseudomonas knee PJI following arthroscopic DAIR with adjunct phage cocktail and antibiotic therapy, achieving clinical resolution on suppressive antibiotic therapy [147]. Tkhilaishvili et al. reported a successful case of recurrent knee PJI, where bacteriophage therapy treated MDR *P. aeruginosa* during repeat TSE in a co-morbid patient with metabolic syndrome and chronic kidney disease [137]. Patey et al. reported a case series of patients treated compassionately with combined bacteriophage and antibiotic therapy over twelve years, including three prosthetic joint infections. Two of these PJI patients achieved infection clearance with partial resolution for the third following phage treatment [148]. Cano et al. treated a case of recurrent knee PJI with targeted intravenous phages in combination with oral antibiotics. Their patient achieved adequate symptom resolution and function, but remained on suppressive oral antibiotics [149].

### 5.2. Ceragenins

Ceragenins or cationic steroid antibiotics (CSA) are a novel antimicrobial class which mimic the activity of endogenous cationic antimicrobial peptides. CSA disrupt bacterial cell membranes through interaction with negatively charged phosphatidylglycerol and lipid A lipopolysaccharides [150]. Activity against a wide range of pathogens including antimicrobial resistant bacteria and yeasts has been demonstrated in vitro including in biofilms and in vivo animal models of urinary tract infection, prosthetic joint infection and peritonitis [151,152,153,154,155]. Furthermore, CSA may be adapted to coat surfaces of medical devices. A clinical trial evaluating the efficacy of ceragenin-coated endotracheal tubes to prevent ventilator associated pneumonia is currently enrolling participants (Ceragenin Coated Endotracheal Tubes for the Prevention of Ventilator Associated Pneumonia (CEASEVAP) [156,157].

### 5.3. Other Novel Options

Antimicrobial peptides are short chains of amino acids with a broad-spectrum, rapid bactericidal action through direct disruption of bacterial cell membranes. Their targeted cell wall disruption allows them to be very effective against biofilm-bacteria during in vitro studies. However, their delivery is a significant issue, with rapid proteolytic degradation and host toxicity currently limiting clinical use, as well as reports of bacterial resistance [158].

Quorum-sensing inhibitors are molecules that disrupt bacterial signalling pathways used to form and maintain the biofilm state and its associated resistance mechanisms. They carry in vitro promise; however, they are currently limited by a narrow therapeutic spectrum as not all species use the same signalling molecules. There are also safety concerns as they may disrupt the signalling of beneficial human microflora, leading to unintended side-effects [158].

Other novel biofilm-disrupting methods include laser phototherapy of implants, bioelectric therapy, and ultrasound microbubble disruption, all of which have shown promise during in vitro or animal studies, but require further development [158].

## 6. Discussion

Prosthetic joint infection remains a major problem for patients, clinicians, and health systems. These infections often require major revision procedures and prolonged antibiotic therapy, and come with significant psychosocial stress and long-term risk of recurrence. They are costly to treat, and healthcare systems only face greater demand in the future. Overcoming the biofilm remains at the core of PJI treatment; however, no ideal solution currently exists. Improving current biomaterials, developing novel carriers, and creating new antimicrobial therapies carry the greatest potential.

The ideal antibiotic carrier for local therapy delivers antibiotics for prolonged duration at effective concentrations, has complete antibiotic release, versatile compatibility with desired antibiotics, and full carrier resorption with minimal risk of allergies or adverse effects systemically or locally. However, all current carrier options are associated with issues, including suboptimal antibiotic delivery or compatibility, carrier persistence, local or systemic complications, or a lack of clinical evidence for their use. Ongoing research into antibiotic carriers including calcium hydroxyapatite, calcium sulphate, bioactive glass and hydrogels, and the development of purpose-built composite biomaterials aim to overcome current problems and develop an ideal antibiotic carrier for use in PJI. However, these novel options require further development in animal models and extensive safety studies before their widespread clinical use.

There are still theoretical shortcomings with local antibiotic therapy, despite its many advantages. Even with much lower systemic antibiotic levels seen in local delivery, there remains a risk of systemic complications such as nephrotoxicity [75]. Local antibiotic therapy should still be seen as an adjuvant to surgical treatment, including thorough debridement, management of dead space, and implant exchange. In the future, despite optimal antibiotic delivery, therapy may still be ineffective due to rising rates of antibiotic resistance. Novel anti-biofilm therapies can overcome this issue with alternate mechanisms of action. Furthermore, they carry greater potential to complement direct antibacterial activity through disruption of the biofilm’s structure. However, these novel therapies, such as phages, require greater development in clinical understanding, product manufacturing, and regulatory approval before they can be relied upon in the clinical setting.

There are several other limitations in our current understanding of local antimicrobial therapy. The most significant are the determinants of local antibiotic effectiveness. This leads to significant heterogeneity in testing methodology across studies and a lack of standardisation of research goals. Reported elution characteristics for various methods are therefore diverse, and it is difficult to evaluate and then successfully translate this into clinical practice. When measuring antibiotic elution, studies report the concentration within local joint fluid, often collected from a post-operative drain. This may misrepresent and reduce effective levels in local tissue [98]. A key advantage of effective local antibiotic therapy is that it may preclude the need for systemic therapy. In the case series by Stockley et al., targeted local antibiotics with TSE have been shown to provide satisfactory success rates without long-term systemic therapy, which can avoid the morbidity and non-adherence of long-term antimicrobial regimes [56]. However, clinical trials to directly compare standard practice to local antibiotic therapy must be performed to validate and quantify this effect. The SOLARIO (Short or Long Antibiotic Regimes in Orthopaedics) trial is currently underway, designed as a non-inferiority trial to compare local antibiotic therapy with under seven days’ systemic therapy to standard systemic therapy of over four weeks [159]. This trial includes PJI patients who have undergone curative revision surgery but excludes DAIR procedures. However, the type of antibiotic, dose, and carrier used are not controlled and are decided by the treating clinicians. Further clinical trials will be required to compare the effectiveness of different antibiotic and carrier combinations. The SOLARIO trial will also help to inform the question posed by the 2018 International Consensus Meeting on Orthopaedic Infections, that concluded there was no evidence to support a beneficial effect of local antimicrobial therapy at that time [160].

In this narrative review we have provided a comprehensive review of local antimicrobial administration options, focused on current clinical outcomes for this difficult patient population to aid clinical practice. We have evaluated antibiotic carriers in bone and joint infection more widely when clinical data in PJI are not available, and we aim to acknowledge literature gaps to guide further clinical and pre-clinical work. 

In summary, local antibiotic therapy has been demonstrated to improve PJI outcomes through a variety of delivery systems. PMMA is the best studied antibiotic carrier in PJI, with the unique ability to form structural spacers that also elute antimicrobials. However, it has suboptimal elution kinetics and is not compatible with all antibiotics. It often requires a second procedure for removal and has been associated with development of antimicrobial resistance due to carrier persistence. Resorbable carriers such as calcium sulphate and hydrogels aim to overcome these issues. Hydrogels are versatile carriers which are compatible with a wide variety of antimicrobial substances, including bacteriophage. They provide reliable elution profiles which may be tailored to meet requirements. Newer solutions such as nanocarriers and anti-biofilm agents aim to deliver antibiotic therapy and eradicate biofilm more effectively; however, these remain at the pre-clinical development stage.

## Figures and Tables

**Figure 1 antibiotics-12-00752-f001:**
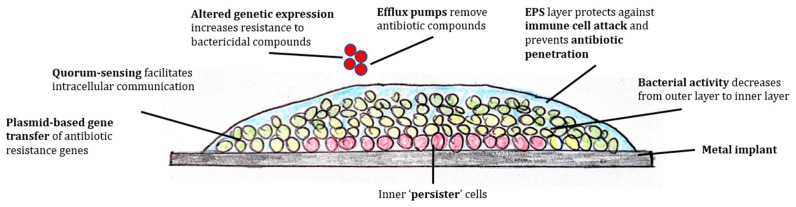
Biofilm resistance to immune clearance and antibiotic therapy.

**Figure 2 antibiotics-12-00752-f002:**
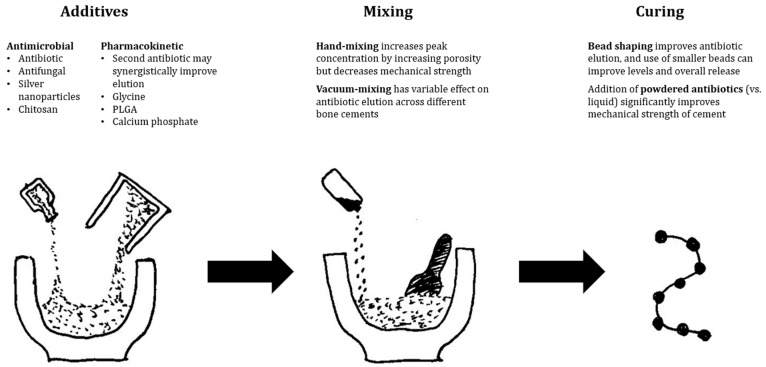
Evidence-based recommendations to improve antibiotic elution from PMMA-based antibiotic delivery vehicles.

**Figure 3 antibiotics-12-00752-f003:**
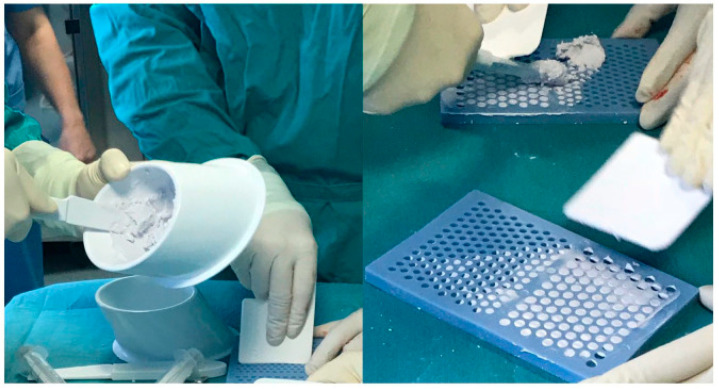
Antibiotic-impregnated Calcium sulphate as cement and shaped into beads (from Ene et al. [92]).

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
