# Peer review of "Local Antibiotic Delivery Options in Prosthetic Joint Infection"

_antibiotics, 2023, doi:10.3390/antibiotics12040752_

Round 1

Reviewer 1 Report

This narrative review focuses on local antibiotics administration to treat osteomyelitis. However, the scope of this article seems to focus more on PMMA and antibiotic filling, which are the major components of local antibiotic delivery in orthopaedic surgery. 

Comments:

On page two, lines 3-5, "Non-antibiotic options target biofilm more efficiently but are still in initial stages of clinical development." Can you give some examples? Also, did the researchers read anything about biofilm-penetrating antibiotics, such as ceragenins, CSA-90 and CSA-131? CSA-131 is a patented and FDA-approved antibiotic coating that has anti-biofilm activities. Also, please add some references to this paragraph.

On page 2, "local antibiotics" section, it is true that local antibiotics have their benefits, but they often require surgery to place and remove the drugs (if beads/ceramics are used). This method has some limitations and disadvantages and should be addressed more comprehensively. Also, this section needs some references.

On page 2, "antibiotic choice" should mention the common use of antibiotics, such as flucloxacillin and cefazolin, clindamycin and gentamicin, and why these antibiotics are used in orthopaedic patients. Antibiotic treatment can harm patients with kidney injury or chronic diseases, but it should also mention that physicians often assess these on a case-by-case basis. Sometimes antibiotics are still used to treat osteomyelitis patients with a lower dose despite their kidney conditions.

On page 6, "... PMMA does not appear to lose mechanical integrity..." Some studies suggest that it reduces, but not much, depending on the concentration of antibiotics (e.g., vancomycin and gentamicin) used. Also, one major limitation of PMMA is the difficulty of controlling the drug release rate. Sometimes, you get a burst release, so the antibiotic concentration is very high for a few hours and then declines, which is also why it has a short half-life in the body. 

On page 7, Table 1, can you summarise the key findings of these studies, and what is the conclusion? Also, what is the relevance to clinical practice?

On page 13, "Bacteriophages' section, bacteriophages have great potential in treating bone and joint infections. However, one limitation is that lytic bacteriophages are challenging to find, and not many bacteriophages that can kill Staphylococcus aureus, which is responsible for more than 50% of osteomyelitis cases. Also, the purification process can be challenging. Nevertheless, bacteriophage therapy is an alternative with a low risk of complications and toxicity in humans.

On page 13, "Other novel option section is very narrow and does not cover all the potential research. Ceragenins (not mentioned) are broad-spectrum antibiotics and have significant applications in orthopaedics. There is also a novel bone-binding antimicrobial (BBA-1) synthesized with bone-targeting and osteogenic properties (not mentioned). There are also bone-resorptive bio-ceramics developed to replace metal implants (not mentioned). 

Additionally, some figures are not original and seem to be copied and pasted from other studies. Creating original figures or summarising the studies' key findings would be better.

Reviewer 2 Report

I sincerely appreciate the opportunity to review the manuscript entitled “Local antibiotic delivery options in prosthetic joint infection” for its publication in Antibiotics.

This review article describes different aspects of antimicrobial treatment of prosthetic joint infection, including administration forms and vehicles, including classical concepts and new developments in the field. Despite its scientific quality and soundness, the authors need to resolve some issues before acceptance.

Major comments

-Considering that this article is a review and expects to be published in a quality and respected journal, the authors should improve the writing of the whole paper. This includes producing an abstract that covers all aspects covered in the text and reporting its major conclusions. Likewise, the conclusions seem cursory and lack adequate academic language, so they also need to be improved. In addition, many paragraphs seem disjointed and interrupt a proper and smooth presentation of the background.

-Authors should be careful in the formatting of the article, including font size and type, margins, missing references, and check the instructions for the "references" section. Sometimes seemed to give the impression that it was not the final version of the article.

-In my opinion, many of the figures included seem to be of no clear use. In addition to not being referenced in the text, I think it would be better for the authors to make their own figures or tables summarizing the major previous findings they want to show and discuss rather than copying the original figure without further discussion.

-I think the authors should include a short paragraph on the prudent use of antimicrobials in this type of pathology, referring to WHO regulations on the use of drugs of different categories of public health relevance. This is necessary so as not to encourage the use of critical drugs, such as vancomycin, when other options are available.

Minor comments

-Minor comments made to the text can be found in the attached pdf file.

Round 2

Reviewer 2 Report

 The comments provided to the first version of the article were adequately considered and included in this new version, thus helping to improve its quality and understanding.

Congratulations.